# Prediction of Chemotherapy Efficacy in Patients with Colorectal Cancer Ovarian Metastases: A Preliminary Study Using Contrast-Enhanced Computed-Tomography-Based Radiomics

**DOI:** 10.3390/diagnostics14010006

**Published:** 2023-12-19

**Authors:** Jinghan Yu, Xiaofen Li, Hanjiang Zeng, Hongkun Yin, Ya Wang, Bo Wang, Meng Qiu, Bing Wu

**Affiliations:** 1Department of Radiology, West China Hospital, Sichuan University, Chengdu 610041, China; fskradio@163.com (J.Y.); zenghanjiang@wchscu.cn (H.Z.); wy58095088@163.com (Y.W.); wangbo_scu@stu.scu.edu.cn (B.W.); 2Department of Medical Oncology, Cancer Center, West China Hospital, Sichuan University, Chengdu 610041, China; lxf0827@163.com; 3Division of Abdominal Tumor Multimodality Treatment, Cancer Center, West China Hospital, Sichuan University, Chengdu 610041, China; 4Institute of Advanced Research, Infervision Medical Technology Co., Ltd., Beijing 100025, China; yhongkun@infervision.com; 5Colorectal Cancer Center, West China Hospital, Sichuan University, Chengdu 610041, China

**Keywords:** colorectal cancer, ovarian metastases, radiomics, chemotherapy efficacy

## Abstract

Ovarian metastasis (OM) from colorectal cancer (CRC) is infrequent and has a poor prognosis. The purpose of this study is to investigate the value of a contrast-enhanced CT-based radiomics model in predicting ovarian metastasis from colorectal cancer outcomes after systemic chemotherapy. A total of 52 ovarian metastatic CRC patients who received first-line systemic chemotherapy were retrospectively included in this study and were categorized into chemo-benefit (C+) and no-chemo-benefit (C−) groups, using Response Criteria in Solid Tumors (RECIST v1.1) as the standard. A total of 1743 radiomics features were extracted from baseline CT, three methods were adopted during the feature selection, and five prediction models were constructed. Receiver operating characteristic (ROC) analysis, calibration analysis, and decision curve analysis (DCA) were used to evaluate the diagnostic performance and clinical utility of each model. Among those machine-learning-based radiomics models, the SVM model showed the best performance on the validation dataset, with AUC, accuracy, sensitivity, and specificity of 0.903 (95% CI, 0.788–0.967), 88.5%, 95.7%, and 82.8%, respectively. All radiomics models exhibited good calibration, and the DCA demonstrated that the SVM model had a higher net benefit than other models across the majority of the range of threshold probabilities. Our findings showed that contrast-enhanced CT-based radiomics models have high discriminating power in predicting the outcome of colorectal cancer ovarian metastases patients receiving chemotherapy.

## 1. Introduction

Colorectal cancer (CRC) is the third primary cause of cancer-related morbidity and mortality worldwide [1]. In women, ovarian metastasis (OM) is associated with high mortality rates, and the median overall survival of patients with OM is 23 months. Furthermore, the 5- and 10-year survival rates are 17% and 8%, respectively [2]. Ovarian metastasis commonly presents as a cystic solid mass, similar to primary ovarian cancer (POC). Hence, ovarian metastasis is occasionally misdiagnosed as POC if the colorectal lesion is subtle. Previous studies have assessed methods for classification using computed tomography (CT) scan or magnetic resonance imaging to help distinguish OM [3,4]. Nevertheless, further clinical validation is required. The survival outcomes of patients improved with systemic chemotherapy, followed by oophorectomy, which is the preferred option for ovarian metastasis from colorectal cancer [5]. However, the ovary is less sensitive to systemic chemotherapy than primary lesions and lesions in other metastatic sites (e.g., the liver and lung). Previous studies have shown that <20% of patients respond well to chemotherapy [6,7]. Hence, it is valuable to anticipate the effectiveness of chemotherapy before treatment to help clinicians develop individualized treatment plans and prevent unnecessary toxicity caused by chemotherapy and the loss of the possible benefits of radical surgery due to disease progression.

Radiomics has promising potential for converting medical images into mineable data and extracting noninvasive virtual characteristics. Significant advancements have been made in lung, breast, and gastric cancers, thereby showing better radiomics values in oncological applications [8,9,10]. However, due to the low incidence and poor prognosis of ovarian metastasis, no studies have been conducted to evaluate and predict its chemotherapeutic response. The primary objective of the present study was to develop a radiomics model utilizing CT scan images in order to construct a histological imaging model. This model was intended to predict the efficacy of chemotherapy in cases of ovarian metastatic colorectal cancer, utilizing pretreatment CT scan image features. Then, internal validation of the model was performed to identify a noninvasive and stable prediction imaging method.

## 2. Materials and Methods

### 2.1. Patients and Study Design

Ethical approval was obtained for this retrospective analysis (2020/1296), and the informed consent requirement was waived. We retrospectively included 68 patients with histologically confirmed ovarian metastases from colorectal cancer from July 2010 to December 2022 at our institution. Inclusion criteria were patients (1) pathologically diagnosed with ovarian metastasis; (2) having received at least two cycles of systemic chemotherapy; (3) with measurable lesions in the ovaries based on the Response Criteria for Solid Tumors 1.1; (4) having complete and available clinical, imaging, and pathological data. The exclusion criteria were as follows: patients with a duration of >1 month from baseline CT scan to chemotherapy, those with incomplete clinical or imaging information, and those with poor-quality CT scan images. In total, 16 patients were excluded due to incomplete clinical data or poor imaging quality, and 52 patients were finally included (Figure 1). Two cycles of chemotherapy were administered to patients in this cohort, and the time interval between baseline and follow-up CT scans was two months. As ovarian metastasis responds poorly to chemotherapy and progresses rapidly, only a few patients achieved partial remission (PR). In this study, stable disease (SD) was defined as disease control, and patients who achieved SD were classified under the chemotherapy-benefit group. Two experienced radiologists evaluated the patient’s baseline and postchemotherapy CT scan images using the Response Criteria for Solid Tumors (RECIST v1.1). Then, the patients were divided into the chemo-benefit group (C+, SD + PR) and the no-chemo-benefit group (C−, PD). Disagreements were resolved via a consensus decision or consultation with a third reviewer.

Clinical data including age, menopausal status, initial tumor location and histological type, status of gene mutations, dimensions and laterality of ovarian metastases, ascites, and other locations for metastasis were collected. In this study, metachronous OM was defined as a time interval of >6 months from primary tumor diagnosis to the discovery of an ovarian metastasis. The initial tumor originated in the right colon (cecum to colonic liver), left colon (spleen to sigmoid), and rectum.

### 2.2. Image Processing and Radiomics Features Extraction

Patients underwent baseline CT scan within 1–3 weeks prior to chemotherapy. During the scan, the patients were requested to suspend their respiration to prevent breathing artifacts. The whole-abdomen CT was performed on patients positioned supine with a slice thickness range of 2–5 mm, and the entire abdomen was scanned (Revolution, GE Healthcare, Milwaukee, WI, USA; Brilliance 64, Phillips, Amsterdam, The Netherlands; SOMATOM, Siemens, Erlangen, Germany). Acquisition and reconstruction parameters were tube current 150–200 mA, tube voltage of 120 kV, pitch 0.8, and matrix size 512 × 512. Section thickness was set at 5 mm and reconstruction section thickness at 1.5 mm. Two experienced diagnostic radiologists performed the segmentation of tumor lesions using ITK-SNAP (version 3.6) software. As the portal lesion was well differentiated from the adjacent tissue, the largest level of the portal lesion was selected for segmentation. Then, segmentation was performed along the border of the tumor, thereby avoiding areas such as the blood vessels and calcifications. If there was a disagreement on segmentation between the two physicians, the two physicians discussed it and reached an agreement.

The CT scan images were initially resampled to a target voxel of 1 mm × 1 mm × 1 mm. Subsequently, the radiomics features were obtained in an automated manner from the manually labeled regions of interest (ROIs) using PyRadiomics software (version 3.0.1), according to the latest recommendations of the image biomarker standardization initiative [11]. Taken together, 1743 radiomic features including 14 shape features, 342 first-order statistics features, and 1387 texture features were extracted from each CT scan with the bin size fixed to 32 [12]. An intraclass correlation coefficient (ICC) analysis was conducted to assess the interobserver reliability of the extracted radiomics features. Radiomics features exhibiting an ICC of 0.75 were considered stable and included in further analysis.

A three-step strategy was used to further select the radiomics features to decrease model complexity and prevent overfitting. Univariate analysis was performed first, and the radiomics features with a significant difference between the C+ and C− groups (Mann–Whitney U test, *p* < 0.05) were kept. Then, the redundant features were removed via Pearson correlation coefficient analysis. If two radiomics features had a high correlation (|r| > 0.95), the feature with a greater *p* value as determined using the Mann–Whitney U test was excluded. Finally, the most critical radiomic features were selected using the least absolute shrinkage and selection operator (LASSO). The penalty parameter was established via 10-fold cross-validation using the “minimum mean-square error” method.

### 2.3. Model Construction and Evaluation

Prior to model development, the values of the identified radiomics features were normalized using z-score normalization. Radiomics models were constructed using five well-known machine learning classifiers, which have been widely used in medical imaging analysis, including logistic regression (LR), naïve Bayes (NBB), random forest (RF), linear discriminant analysis (LDA), and support vector machine (SVM). Due to the limited sample size in this study, the predictive models were constructed and validated via leave-one-out cross-validation. To mitigate the issue of class imbalance, the balanced weight strategy was implemented. This meant adjusting the weights of the chemo-benefit and no-chemo-benefit classes in inverse proportionality to their respective prevalences.

The model performance was evaluated via receiver operating characteristics analysis with respect to the area under the receiver operating characteristic curve (AUC). Furthermore, in accordance with the maximal Youden index criteria, the accuracy, sensitivity, specificity, positive predictive value (PPV), and negative predictive value (NPV) were computed using the optimal threshold. The construction and evaluation of the radiomics models were performed with the InferScholar platform (InferVision ver3.5).

### 2.4. Calibration and Decision Curve Analysis

The calibration of the predictive models was examined via 1000 bootstrapping resampling, and the consistency between the actual observed rate and the predicted probability was evaluated using the Hosmer–Lemeshow test [13]. To compare the clinical utility of the predictive models, decision curve analysis (DCA) was additionally conducted by calculating the net benefits for a range of threshold probabilities [14].

### 2.5. Statistical Analysis

Regarding the clinical characteristics, frequencies and percentages are used to express descriptive data and mean ± standard deviation to represent parametric variables. Categorical variables were analyzed using Pearson’s chi-square test or Fisher’s exact test. Continuous variables were compared using the Mann–Whitney U test or Student’s *t*-test. All statistical analyses were performed using the Statistical Package for the Social Sciences software (version 27.0). A *p* value of <0.05 was considered statistically significant.

## 3. Results

### 3.1. Patients’ Characteristics

Finally, 52 patients were included in this study. Table 1 shows the clinicopathologic characteristics of all patients. Changes in the size of the ovarian metastatic tumors of each patient were determined according to the RECIST criteria after chemotherapy. In total, 25 patients who achieved SD and PR were classified in the C+ group, while, 27 patients who achieved PD were included in the C− group. The median age at ovarian metastasis diagnosis was 46 (range: 25–77) years. In total, 55.8% (*n* = 29) of patients were premenopausal. The mean carcinoembryonic antigen (CEA) and cancer antigen (CA-125) levels upon OM diagnosis were 116.1 (normal range: 0–5) ng/mL and 87.2 (normal range: 0–25) U/mL, respectively.

Based on contrast-enhanced CT scan images, ovarian metastasis was classified as solid and cystic in 19 (36.5%) patients, multicystic with or without nodules in 30 (57.6%), and cystic in 3 (5.8%). There were two (3.8%) patients with solitary OM. Among the 52 patients in this study, 30 (57.7%) cases combined with liver metastasis, 23 (44.2%) cases combined with peritoneal metastasis, and 7 (13.5%) cases combined with lung metastasis. Furthermore, 16 (30.8%) patients presented with metastasis to sites such as the bone, brain, and nonregional lymph nodes. In total, 52 patients received systemic chemotherapy with or without targeted drugs (FOLFIRI, *n* = 17 (32.7%); FOLFOX, *n* = 16 (30.7%); and XELOX, *n* = 19 (36.5%)).

### 3.2. Radiomics Features’ Extraction and Validation

#### 3.2.1. Selection of Radiomics Features

After ICC analysis, univariate analysis, and Pearson correlation analysis, 162 radiomics features were retained in LASSO regression analysis. Finally, five radiomics features with nonzero coefficients were selected for model construction under the optimal tuning parameter lambda (Figure 2). Figure 3 shows the heatmap of the selected radiomics features with standardized values.

#### 3.2.2. Model Performance

A comparative analysis of the discrimination performance of the predictive radiomics models on the training and validation datasets was conducted using receiver operating characteristic curve analysis (Figure 4). The AUCs of the NBB model, LDA model, LR model, RF model, and SVM model were 0.901 (95% confidence interval (CI): 0.786–0.966), 0.900 (95% CI: 0.784–0.966), 0.898 (95% CI: 0.782–0.965), 0.949 (95% CI: 0.850–0.991), and 0.892 (95% CI: 0.775–0.961) on the training dataset, and 0.823 (95% CI: 0.692–0.915), 0.852 (95% CI: 0.726–0.935), 0.883 (95% CI: 0.764–0.955), 0.792 (95% CI: 0.656–0.892), and 0.903 (95% CI: 0.778–0.967) on the validation dataset, respectively. In the training dataset, no significant differences were found among the radiomics models (all *p* values > 0.05), and the AUC of the SVM model was significantly higher than that of the NBB model (*p* = 0.024) and the LDA model (*p* = 0.045) on the validation dataset. Among these machine learning classifier-based radiomic models, only the RF model had a tendency toward overfitting, as its AUC was considerably higher on the training dataset than on the validation dataset (*p* = 0.026). In addition, the accuracy, sensitivity, specificity, PPV, and NPV of these models under the optimal cutoff point in the validation dataset are shown in Table 2.

#### 3.2.3. Clinical Utilities

On the validation dataset, the clinical utility of the five radiomics models was assessed through calibration analysis and DCA. All models had good calibration as the nonsignificant statistics of the NBB model, LDA model, LR model, RF model, and SVM model were 0.472, 0.588, 0.098, 0.110, and 0.125 on the validation dataset, respectively (Figure 5). In addition, DCA showed that all radiomics models outperformed both the treat-all and treat-none approaches. The SVM model outperformed alternative models in terms of net benefit for the majority of threshold probabilities (Figure 6).

## 4. Discussion

CRC is one of the cancers with the highest incidence and mortality rate in the 21st century, with the second (9.4%) and third (9.5%) highest incidence and mortality rates among all malignancies in female patients [15]. Approximately 3–5% of female patients with CRC develop ovarian metastases, and this group of patients commonly has a poor prognosis, with a median survival of 13–36 months based on previous studies [5,16,17]. Some patients with ovarian metastases present with nonspecific clinical signs such as abdominal distension, abdominal pain, and anemia [18]. Ovarian metastasis is morphologically similar to POC, and 70% of patients with OM have high CA125 levels. Therefore, ovarian metastasis with an insidious primary tumor is occasionally misdiagnosed as POC [19,20]. There is a lack of large-sample prospective trials on the preferred treatment strategy due to the low incidence and poor prognosis of ovarian metastasis. In this context, this study was conducted to improve colorectal cancer ovarian metastasis management by evaluating OM-related imaging features with the combined analysis of clinicopathological features and treatment strategies.

The main treatment options for ovarian metastasis include oophorectomy, cytoreductive surgery, hyperthermic intraperitoneal chemotherapy, and systemic therapy. Although metastatic colorectal cancer is predominantly treated with systemic therapy, previous reports have revealed that ovarian metastases exhibit a notably lower sensitivity to systemic chemotherapy in comparison to metastases in other organs, with an objective response rate of <20% [6,21]. Some retrospective studies have shown that in patients with ovarian metastasis, the benefit of oophorectomy is significant, with a prolonged median ovary-specific survival (date of ovarian metastasis diagnosis to death, 20.8 months vs. 10.9 months) and progression-free survival (15.6 months vs. 6.1 months) compared with those who did not undergo oophorectomy [7]. Other studies reported a significant improvement in survival in patients with complete cytoreduction surgery (5-year survival rate of 47%; median survival of 48 months) [22]. The achievement of complete cytoreduction was considered an independent predictor of improved prognosis [23,24]. Some studies recommend a prophylactic bilateral ovariectomy because if one ovary proves to be metastatic, there is an equal probability that the other ovary will also be invaded and may have developed microscopic metastases [25,26]. However, current studies are small-scale retrospective studies, and there are no large-scale prospective studies confirming the prognostic improvement effect of oophorectomy in patients with colorectal cancer. By contrast, in women who are premenopausal or have reproductive needs, oophorectomy may have a deleterious influence on their hormonal and psychological health [27]. Furthermore, surgery may lead to organ damage or secondary surgical adhesions. Therefore, an in-depth assessment of the risks and benefits associated with the treatment modality is required.

Thus far, the mechanism of resistance to chemotherapeutic drugs in ovarian metastasis is not clear. With the development of sequencing technology, tumor-related genes are being increasingly identified, and RAS, BRAF, and PIK3CA mutations are being detected in colorectal cancer. In individuals with chemotherapy-resistant colorectal cancer, RAS mutations increased the risk of ovarian metastases (HR = 3.12) [28]. Previous studies have shown that combined treatment with bevacizumab and fluorouracil, irinotecan, leucovorin, and oxaliplatin improves the prognosis of metastatic colorectal cancer [29,30]. Cetuximab, as a first-line treatment, reduced the risk for patients with metastatic colorectal cancer who exhibited wild-type progression of KRAS [31]. In our study cohort, 18 (34.6%) patients developed RAS gene mutations; 1 (1.9%) developed BRAF gene mutation; and 1 developed PIK3CA gene mutation. No statistically significant distinction was observed between the C+ and C− groups with regard to the administration of targeted drug therapy or chemotherapy regimen. This could be attributed to the fact that genetic status testing was not conducted for all patients in the study, and genetic status monitoring was not carried out as the patients’ diseases advanced. Thus far, there are no reliable evaluation tools or predictors in patients who benefit from chemotherapy. Systemic chemotherapy is the preferred regimen for patients who are medically unfit or unwilling to undergo surgery.

Radiomics aims to extract an extensive array of quantitative features from clinical images using data feature algorithms that can uncover disease characteristics. Radiomics can predict tumor genotype, microsatellite stability status, and prognosis preoperatively [32,33,34]. Only a few studies have investigated the efficacy of systemic chemotherapy for patients with ovarian metastases from colorectal cancer. Conventional imaging has an extremely limited capacity for prediction. Therefore, the current study primarily aimed to establish an accurate prognostic prediction model for ovarian metastasis from colorectal cancer chemotherapy response utilizing a CT-scan-based radiomics model and machine learning. Five models of conventional machine learning models were employed in order to assess and validate the accuracy of each predictive model. The SVM model had a significantly higher AUC on the validation set (0.903, 95% CI: 0.788–0.967), with a sensitivity and specificity of 95.7% and 82.8%, respectively. The remaining four models had an area under the curve of >0.7, which indicated a good predictive value and potential for further application. Therefore, our CT-scan-based radiomics model could be beneficial as an early predictor of the efficacy of chemotherapy against ovarian metastases and identify patients who are primary-drug-resistant and who do not respond well to chemotherapy, thereby allowing for a more personalized treatment plan such as chemotherapy combined with surgery, interventional therapy, and radiotherapy. In the no-chemotherapy-benefit group, it is important to reduce the toxicity caused by chemotherapeutic agents and select radical treatment before disease progression. In the chemotherapy-benefit group, chemotherapy combined with oophorectomy may prevent trauma associated with subtractive surgery, such as vascular, organ, or nerve damage.

Several limitations exist in the present study. First, it was retrospective in nature, and confounding variables such as chemotherapy regimen, cycle time, and postimaging modalities may have caused bias. Second, the study’s sample size was small, and an imbalance in sample size between groups could have impacted the predictive model’s performance. Therefore, prospective studies with larger sample sizes should be performed to verify the study findings. Third, data were obtained from only one medical institution, which might limit the generalizability of the model. Further independent validation sets are required to confirm our findings. Fourth, deep learning was not used to build the model in this study, and we did not apply joint modeling of pathological features. By integrating deep learning or pathological features with CT scan imaging histology, the accuracy of prognosticating the effectiveness of systemic chemotherapy for ovarian metastatic colorectal cancer can be further enhanced. This then reduces the error rate of the prediction model and increases the confidence level.

## 5. Conclusions

Five prediction models for evaluating systemic chemotherapy in patients with ovarian metastasis from colorectal cancer were established using machine learning algorithms. Among them, the SVM model had the best predictive ability, with strong differentiation on both the test and validation sets, and had high sensitivity, specificity, and accuracy. We proposed the use of a CT-scan-based radiomics baseline model for distinguishing potential responders from nonresponders after systemic chemotherapy for ovarian metastases in colorectal cancer. Radiomics profiling can be used as an adjunct to clinical treatment decision making to help oncologists predict chemotherapy response, leading to timely and effective individualized treatment planning for potential nonresponders. We hope that the study results can provide a basis for large-scale cohort studies. Nevertheless, prospective studies should be performed to improve the individualized prediction of responders in patients with ovarian metastatic colorectal cancer.

## Figures and Tables

**Figure 1 diagnostics-14-00006-f001:**
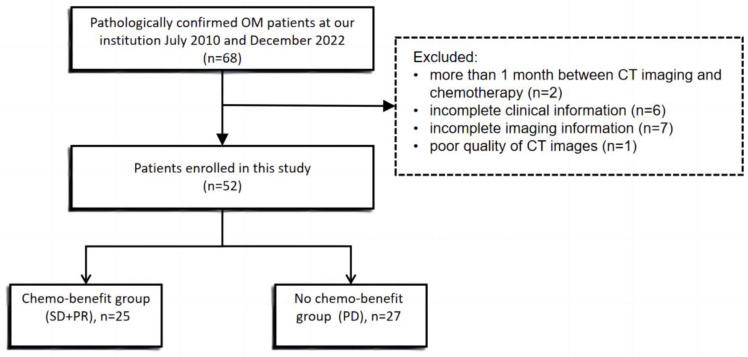
Work flowchart of patient selection. OM, ovarian metastasis; SD, stable disease; PR, partial response; PD, progressive disease.

**Figure 2 diagnostics-14-00006-f002:**
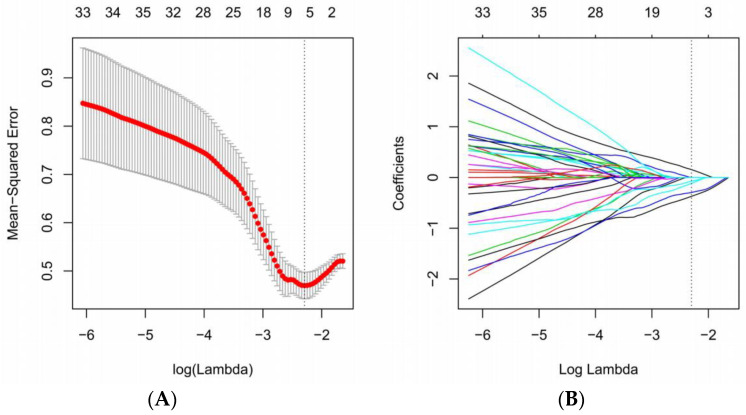
Radiomics feature selection via LASSO regression analysis. (**A**) Utilization of 10-fold cross-validation with the minimal mean squared error criteria to choose the optimal tuning parameter lambda. (**B**) The coefficient profile plot of five nonzero coefficients against the optimal log (lambda) sequence.

**Figure 3 diagnostics-14-00006-f003:**
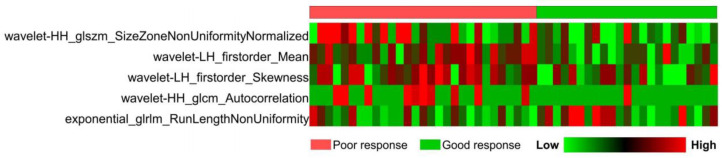
Heatmap of the selected radiomics features.

**Figure 4 diagnostics-14-00006-f004:**
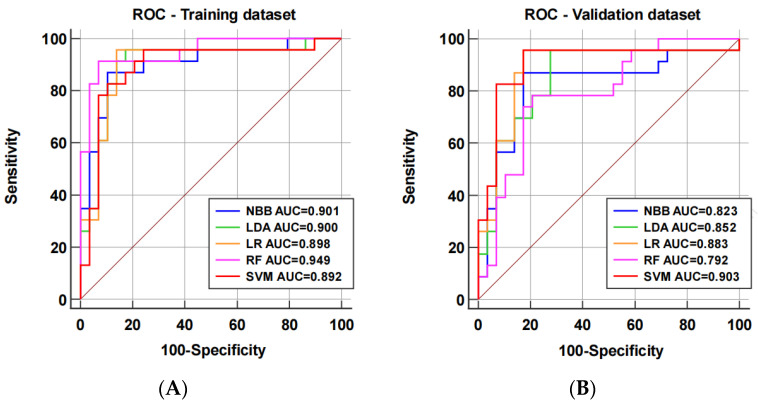
ROC analysis of the predictive models on the training dataset (**A**) and the validation dataset (**B**).

**Figure 5 diagnostics-14-00006-f005:**
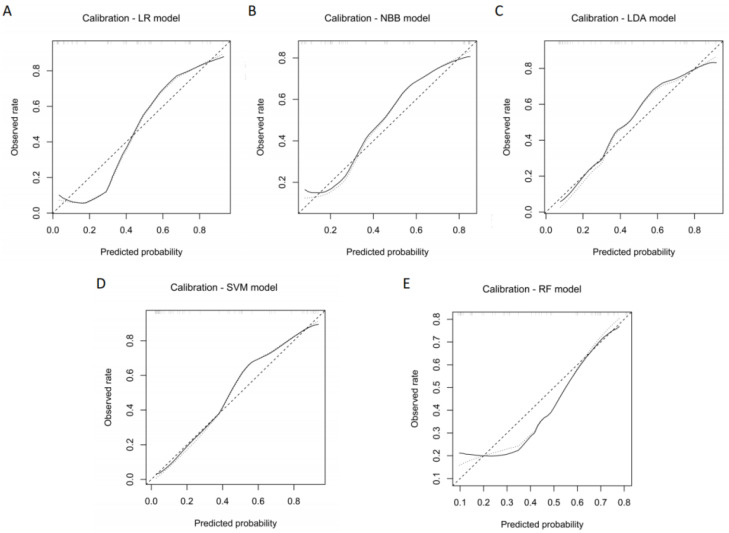
Calibration analysis of the LR model (**A**), NBB model (**B**), LDA model (**C**), SVM model (**D**), and RF model (**E**) on the validation dataset.

**Figure 6 diagnostics-14-00006-f006:**
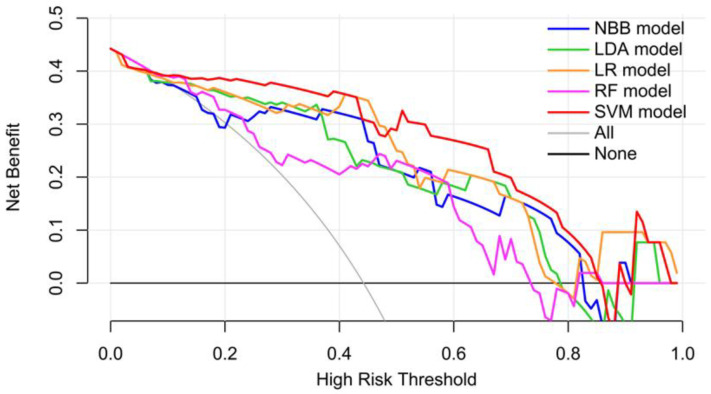
Analysis of the predictive models’ decision curves on the validation dataset.

**Table 1 diagnostics-14-00006-t001:** Clinicopathological characteristics of all the enrolled patients.

Characteristics of Patients and Primary Tumors	Total (*n* = 52), *n* (%)	Chemo-Benefit Group (*n* = 25), *n* (%)	No-Chemo-Benefit Group (*n* = 27), *n* (%)	*p* Value
Age at diagnosis of OM (years, mean ± SD)	46.0 ± 11.7	47.1 ± 13.6	45.0 ± 9.8	0.58
Menopausal status				0.40
Premenopausal	29 (55.8)	13 (52.0)	16 (59.3)	
Postmenopausal	23 (44.2)	12 (48.0)	11 (40.7)	
Primary tumor location				0.58
proximal colon	12 (23.0)	7 (28.0)	5 (18.5)	
distal colon	20 (38.5)	8 (32.0)	12 (44.4)	
rectum	20 (38.5)	10 (40.0)	10 (37.0)	
Histological type of primary tumor				0.18
Adenocarcinoma	37 (71.2)	16 (64.0)	21 (77.8)	
Mucinous adenocarcinoma	13 (25)	8 (32.0)	5 (18.5)	
Signet ring cell carcinoma	2 (3.8)	1 (4.0)	1 (3.7)	
T stage of primary tumor				0.95
T2	7 (13.5)	3 (12.0)	4 (14.8)	
T3	10 (19.2)	5(20.0)	5 (18.5)	
T4	35 (67.3)	17 (68.0)	18 (66.7)	
N stage of primary tumor				0.38
N0	4 (7.7)	3 (12.0)	1 (3.7)	
N1	21 (40.4)	11 (44.0)	10 (37.0)	
N2	27 (51.9)	11 (44.0)	16 (59.3)	
Distal metastasis				0.23
M1a	2 (3.8)	2 (8.0)	0	
M1b	27 (51.9)	14 (56.0)	13 (48.1)	
M1c	23 (44.2)	9 (36.0)	14 (51.8)	
Ascites				0.48
Yes	7 (13.5)	2 (8.0)	5 (18.5)	
No	45 (86.5)	23 (92.0)	22 (81.5)	
CEA (ng/mL, mean ± SD)	116.1 ± 212.7	26.7 ± 20.8	198.8 ± 271.2	0.008
CA125 (U/mL, mean ± SD)	87.2 ± 171.6	47.9 ± 42.3	123.5 ± 230.7	0.053
CA19-9 (U/mL, mean ± SD)	331.0 ± 524.4	393.1 ± 695.1	273.5 ± 293.6	0.86
RAS mutation				0.84
Mutant	18 (34.6)	9 (36.0)	9 (33.3)	
BRAF mutation				0.51
Mutant	1 (1.9)	0	1 (3.7)	
PIK3CA mutation				0.51
Mutant	1 (1.9)	0	1 (3.7)	
Timing of metastases				0.60
Synchronous	22 (42.3)	9 (36.0)	13 (48.1)	
Metachronous	30 (57.7)	16 (64.0)	14 (51.9)	
Ovarian metastases laterality				0.79
Bilateral	24 (46.2)	12 (48.0)	12 (44.4)	
Unilateral	28 (53.8)	13 (52.0)	15 (55.6)	
Longest diameter of OM (cm, mean ± SD)	5.9 ± 3.9	5.5 ± 3.6	6.2 ± 4.2	0.43
Morphology				0.68
Solid and cystic	19 (36.5)	10 (40.0)	9(33.3)	
Multicystic without nodules	25 (48.1)	10 (40.0)	15 (55.6)	
Multicystic with nodules	5 (9.6)	3 (12.0)	2 (7.4)	
Cystic	3 (5.8)	2 (8.0)	1 (3.7)	
Chemotherapy regimens				0.77
XELOX	19 (36.5)	10 (40.0)	9 (33.3)	
FOLFOX	16 (30.7)	7 (28.0)	9 (33.3)	
FOLFIRI	17 (32.7)	8 (32.0)	9 (33.3)	
Targeted Drugs				0.84
No	17 (32.7)	8 (32.0)	9 (33.3)	
Bevacizumab	23 (44.2)	12 (48.0)	11 (40.7)	
Cetuximab	12 (23.1)	5 (20.0)	7 (25.9)	

OM, ovarian metastasis; CEA, carcinoembryonic antigen; CA125, carbonhydrate antigen 125; CA19-9, carbonhydrate antigen 19-9; XELOX (oxaliplatin, capecitabine); FOLFOX (oxaliplatin, calcium folinate, fluorouracil); FOLFIRI (irinotecan, calcium folinate, fluorouracil).

**Table 2 diagnostics-14-00006-t002:** Detailed performance of the NBB model, LDA model, LR model, RF model, and SVM model on the validation dataset.

Model	AUC (95% CI)	*p* Value	Accuracy (%)	Sensitivity (%)	Specificity (%)	PPV (%)	NPV (%)
NBB	0.823 (0.692–0.915)	0.024	84.6	87.0	82.8	80.0	88.9
LDA	0.852 (0.726–0.935)	0.045	82.7	95.7	72.4	73.3	95.5
LR	0.883 (0.764–0.955)	0.282	88.5	95.7	82.8	81.5	96.0
RF	0.792 (0.656–0.892)	0.051	78.8	78.3	79.3	75.0	82.1
SVM	0.903 (0.788–0.967)	reference	88.5	95.7	82.8	81.5	96.0

## Data Availability

The data presented in this study are available on request from the corresponding author.

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
