# Peer review of "Prediction of Chemotherapy Efficacy in Patients with Colorectal Cancer Ovarian Metastases: A Preliminary Study Using Contrast-Enhanced Computed-Tomography-Based Radiomics"

_diagnostics, 2023, doi:10.3390/diagnostics14010006_

Round 1

Reviewer 1 Report

Comments and Suggestions for Authors

The manuscript aims to predict treatment respons of chemotherapy on ovarian metastasis in patients with colorectal cancer using radiomics.

The manuscript is overall well written and the evaluation is extensive. I have not major concerns on the study design. The clinical relevance, however, seems limited from the small cohort, strict inclusion criteria, limited follow-up, and the fact that patients with ovarian metastasis often receive surgery anyhow. Despite this, the manuscript is a relevant contribution to the literature to demonstrate that this radiomics prediction models are feasible. 

My concerns and comments are (not in arbitrary order):

- Please make more clear in the text that you aim to predict treatment respons based on RECIST after two cycles of systemic therapy. 

- The methods could be improved by including clinical parameters of pathology features in the model. Consider to add to the methods of discussion.

- Please explain when patients with suspected ovarian metastasis receive surgery in your institute. Did patient with excellent treatment respons not undergo surgery? The may be a bias.

- In figure 1, please add the number of patients for the all boxes and all exclusion criteria.

- I would prefer not the abbreviate ovarian metastasis for readability.

- In the introduction, first paragraph: please change ...previous studies have assessed the optimal method, to a method. And Hence, it is crucial, to Hence, it is valuable (or a synonym).

 - Under the methods, paragraph 2.2, please explain what a large dispute was between the observers. Change dispute to disagreement.  And late, please explain the fixed bin size to 32. This seems a parameter, but will not be clear to all readers.

- The discussion is too extensive and should be limited to the scoop of this manuscript. Do not repeat exact numbers of the methods in the discussion.  

Manuscript may benefit from a  statistical reviewer. 

Comments on the Quality of English Language

Well written. No major concerns. 

Author Response

Comment 1 Please make more clear in the text that you aim to predict treatment respons based on RECIST after two cycles of systemic therapy.

Response 1 We are grateful for your careful review and thank you for your thoughtful recommendations. Two cycles of chemotherapy were administered to patients in this cohort, and the time interval between baseline and follow-up CT scans was two months. We aimed to evaluate the change in ovarian metastases at follow-up CT versus baseline CT for early chemotherapy efficacy analysis. We have added this to the method (Page 2, Line 78-80).

Comment 2 The methods could be improved by including clinical parameters of pathology features in the model. Consider to add to the methods of discussion.

Response 2 Thank you for your comments and suggestions. We are considering the next step of joint modeling of pathohistological features with the radiomic features, as well as further combining deep learning to improve the accuracy and applicability of the model. We have added this to the limitation (Page 11, Line 339-340).

Comment 3 Please explain when patients with suspected ovarian metastasis receive surgery in your institute. Did patient with excellent treatment response not undergo surgery? The may be a bias.

Response 3 Thank you for your careful review and questions. We usually give surgical recommendations for patients with low staging and good response to chemotherapy, while some patients opting for surgery and some opting for maintenance chemotherapy. It is available to add other options of treatment after systemic chemotherapy, depending on the patient's age, surgical tolerance, and financial factors. In our study, the assessment of the chemotherapy effectiveness is conducted before surgery or other treatment options.

Comment 4 In figure 1, please add the number of patients for the all boxes and all exclusion criteria.

Response 4 Thank you for pointing this out. We have made correction in figure 1(page 3, first line) according to your suggestion, and added reasons for excluding patients and specific numbers.

Comment 5 I would prefer not the abbreviate ovarian metastasis for readability.

Response 5 We are grateful for your recommendations. Since the imprecise definition of CRCOM, this expression has been deleted from the text, and OM usage has been modified properly.

Comment 6 In the introduction, first paragraph: please change ...previous studies have assessed the optimal method, to a method. And Hence, it is crucial, to Hence, it is valuable (or a synonym).

Response 6 Thank you for your careful review and suggestions. Page 2, line 46, the statements “the optimal method” were corrected as “methods”. Page 2, line 52, the statements “crucial” were corrected as “valuable”.

Comment 7 Under the methods, paragraph 2.2, please explain what a large dispute was between the observers. Change dispute to disagreement.  And late, please explain the fixed bin size to 32. This seems a parameter, but will not be clear to all readers.

Response 7 We are very sorry for our incorrect writing. In our manuscript, “large dispute” was different opinion in segmentation. Page 3, line 110, the statement “large dispute” was corrected as “disagreement”. In page3, line 121, “the fixed bin size to 32” means a fixed bin count and a fixed bin width, which is to ensure that the fineness of the histograms is consistent across all images, ensuring that the texture features are informative and comparable between lesions. We have inserted new literature in the paragraph (page 3, line 118) and updated the order of the other literature in the text.

Comment 8 The discussion is too extensive and should be limited to the scoop of this manuscript. Do not repeat exact numbers of the methods in the discussion.

Response 8 We are grateful for your careful review and thank you for your thoughtful recommendations. We have adjusted and revised this part according to the reviewer’s suggestion.

Reviewer 2 Report

Comments and Suggestions for Authors

The authors explore the use of contrast-enhanced CT-based radiomics models in predicting outcomes for ovarian metastasis from colorectal cancer after chemotherapy using baseline CT scans (pre-treatment). The researchers retrospectively analyzed 52 patients, categorizing them into chemo-benefit (C+, SD+PR) and no chemo-benefit (C-, PD) groups. Various radiomics models were constructed and evaluated, with the SVM model demonstrating the best performance in terms of AUC, accuracy, sensitivity, and specificity. The results indicate that these radiomics models possess significant discriminatory capability in forecasting outcomes for colorectal cancer patients with ovarian metastases undergoing chemotherapy.

The work seems to be an interesting preliminary study using radiomics to predict the outcome of colorectal cancer ovarian metastases, but some aspects should be explained.

CRCOM is not explicitly defined. I would suggest inserting a list of abbreviations.

At the end of the Introduction paragraph, the authors state: "Then, external validation of the model was performed to identify a noninvasive and stable prediction". I would have expected the authors to present an external validation in their work. The external validation involves assessing a model's performance on an independent dataset not used during training, providing a more robust evaluation of its generalization capabilities. If external validation is conducted, it would typically be mentioned separately in the methodology or results section of the study. However, the Materials and Methods paragraph mentioned that Leave-One-Out Cross-Validation was used because the dataset was small and imbalanced. The Leave-One-Out Cross-Validation can be a reasonable choice in this case (since the chosen metrics are sensitive to the performance of the minority class), but it is not an external validation. The author should clarify and correct this aspect.

The analyzed images were acquired from 3 different scans, but they did not discuss any harmonization strategy among different acquisition protocols.

The result generalizability is poor due to the lack of external validation and the monocentric study design. The authors underlined this fact in the discussion of the limitations of this study.

Author Response

Comment 1 CRCOM is not explicitly defined. I would suggest inserting a list of abbreviations.

Response: We are grateful for your recommendations. Since the imprecise definition of CRCOM, this expression has been deleted from the text, and OM usage has been modified properly.

Comment 2 At the end of the Introduction paragraph, the authors state: "Then, external validation of the model was performed to identify a noninvasive and stable prediction". I would have expected the authors to present an external validation in their work. The external validation involves assessing a model's performance on an independent dataset not used during training, providing a more robust evaluation of its generalization capabilities. If external validation is conducted, it would typically be mentioned separately in the methodology or results section of the study. However, the Materials and Methods paragraph mentioned that Leave-One-Out Cross-Validation was used because the dataset was small and imbalanced. The Leave-One-Out Cross-Validation can be a reasonable choice in this case (since the chosen metrics are sensitive to the performance of the minority class), but it is not an external validation. The author should clarify and correct this aspect.

Response 2 We apologize for the inaccurate statement in the introduction. External validation has not been performed in our study and we have made the correct change in the statement (Page 2, line 64).

Comment 3 The analyzed images were acquired from 3 different scans, but they did not discuss any harmonization strategy among different acquisition protocols.

Response 3 We are grateful for your suggestion. The coordination strategy between three scans were added (Page3, line 103-105).

Comment 4 The result generalizability is poor due to the lack of external validation and the monocentric study design. The authors underlined this fact in the discussion of the limitations of this study.

Response 4 The sample size of this study was limited due to the low incidence and poor prognosis of colorectal cancer ovarian metastasis. We consider combining deep learning or joint models in the future, or conducting multi-center studies with larger sample sizes.
